# Measuring Offline Robustness in Reinforcement Learning

## Abstract

Recent work in reinforcement learning has focused on several characteristics of learned policies that go beyond maximizing reward. These properties include fairness, explainability, generalization, and robustness. In this paper, we define *offline robustness* (OR), a measure of how much variability is introduced into learned policies by incidental aspects of the training procedure, such as the order of training data or the particular exploratory actions taken by agents. A training procedure has high OR when the agents it produces take very similar actions on a set of offline test data, despite variation in these incidental aspects of the training procedure. We develop an intuitive, quantitative measure of OR and calculate it for eight algorithms in three Atari environments across dozens of interventions and states. From these experiments, we find that OR varies with the amount of training and type of algorithm and that high performance does not imply high OR, as one might expect.

## 1 Introduction

Modern AI systems make inferences that are complex, useful, and often incomprehensible. This is particularly true of AI systems that use deep neural networks. Such systems perform well on many types of tasks, but their inferences are largely or completely incomprehensible to users (Olson et al., 2021; Atrey et al., 2020; Loyola-Gonzalez, 2019; Witty et al., 2021; Doshi-Velez & Kim, 2017). Prospective users of deployed AI systems have little or no information about when they ought to trust a system's output, and when to discount that output as erroneous. This is particularly problematic when users must decide whether to delegate some degree of control to the system (e.g., autonomous driving).

Research in *explainable artificial intelligence* (XAI) is intended to address this problem. XAI aims to develop methods for understanding, appropriately trusting, and effectively managing AI systems Gunning et al. (2019). To date, most research in XAI has focused on supervised learning tasks, such as classification and regression (Došilović et al., 2018), rather than on the sequential decision-making tasks typically addressed by reinforcement learning (RL) and planning. This is somewhat surprising, given that sequential decision-making is often both high-stakes (as in autonomous driving, medical treatment, and industrial process control) and challenging to explain (Rudin, 2019). As a result, there is a growing literature on *explainable reinforcement learning* (XRL), and the number and variety of methods in XRL is expanding rapidly (Puiutta & Veith, 2020; Coppens et al., 2019; Madumal et al., 2020; Verma et al., 2018; Rusu et al., 2016; Khan et al., 2009; Elizalde et al., 2008).

Explainable reinforcement learning can have several goals. Perhaps the simplest goal is to explain the individual decisions of a reinforcement learning agent, such as why the agent took an action in a specific context. Explanations of individual low-level actions are useful in some situations, such as the post-hoc analysis of accidents. However, explaining the broader, high-level behavior of the system that produced the RL agent is a more widely applicable goal. Such explanations would help users construct accurate mental models of how agents make decisions and act in general, and this would allow users to invest appropriate levels of trust in such systems. For example, such broad explanations might help a human user decide when to delegate to an agent or deploy the agent to operate autonomously Gunning et al. (2019); Druce et al. (2019); Holzinger et al. (2019). Importantly, RL data sets or environments may change over time, so an

agent's explanations should support some degree of generalization. Helping users understand these systems is a lofty goal, and it may be impossible, particularly for complex models such as deep networks (Rudin, 2019). Appropriately defining the high-level behavior of a deep RL agent requires human interpretation and is specific to the environment and the available set of actions. Therefore, we restrict our scope to providing a quantitative measure that may be useful when examining the behavior of deep RL agents.

In this work, we formally define a property of systems produced by reinforcement learning: the *robustness* of RL agents. We refer to the procedure used for training an RL agent as an *RL training pipeline* or, when clear from context, simply as a *pipeline*. A generic pipeline defines a distribution over pipeline instances, where the variability is due to incidental aspects of the training procedure (e.g., the random seed or order of the training data). A *pipeline instance* is produced by instantiating these aspects of a generic pipeline and encompasses all the components that produce the trained agent, including those aspects common to all instances in the distribution (e.g., the algorithm and architecture used for training), as well as those aspects that vary (e.g., the order of the training data, network weight initialization, random seed, and other arbitrary choices). Such choices can be thought of as random draws from a large population of valid options (e.g., random seeds or training sets). A specific policy is produced by running a pipeline instance, and if that pipeline instance were run again, precisely the same policy would be produced. A given RL training pipeline is capable of producing many different pipeline instances and thus many different policies. Thus, a pipeline produces a distribution over policies. Hereafter, we refer to the agent executing the policy produced by a pipeline instance as the *RL agent* or simply *agent*. Consequently, every time an agent is retrained or updated and redeployed, this constitutes a new draw from the generic pipeline's policy distribution. In this paper, we define *offline robustness* (OR) as a property of a distribution of policies produced by an RL training pipeline. A pipeline produces robust policies when the policies in that distribution behave similarly in a variety of situations.

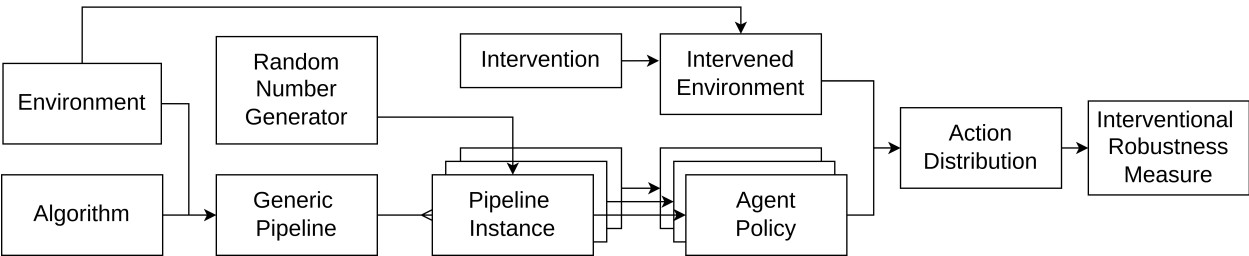

Figure 1: **RL Training Pipelines.** In this work, we characterize RL pipelines according to this diagram. The environment and RL training algorithm form a *generic pipeline*, which can be instantiated into a *pipeline instance*, where the random seed and all incidental training features are fixed. A generic pipeline can produce many pipeline instances, but each pipeline instance only produces one agent policy. Using these policies in the environment under intervention, we calculate the OR metric $\mathcal{R}$ from sampled action distributions according to Algorithm 2.

OR is the degree to which a given pipeline produces policies that generate similar actions for a set of offline test states, despite incidental differences in the training procedure. For example, consider a deep RL training pipeline applied to a highly regular domain, such as Atari games. Such a pipeline can produce policies that memorize a sequence of actions rather than learning a more generalized policy (Zhang et al., 2018). Changing the random network initializations or the order in which training data is presented can produce policies that take identical actions in familiar, on-policy states, but radically different actions in unfamiliar, out-of-distribution states (Witty et al., 2021). In our terminology, such a pipeline (and its resulting policies) is not robust.

It is, of course, possible that the policies produced by a set of pipeline instances could be non-robust at the level of individual actions, but still robust in terms of higher-level strategies. In such a case, different policies would implement the same high-level strategy via different low-level action sequences. OR can also be used with macro-actions, such as options, to measure robustness at higher levels (Sutton et al., 1999).

Not all modern RL pipelines are robust, and this can be acceptable when optimizing for on-policy performance. Indeed, low robustness may be desirable if a system designer wants agents that employ a wide variety of strategies. On the other hand, robust pipelines provide several benefits. First, system engineering is far simpler and more reliable when RL pipelines are robust. Over the lifetime of the system, the design and implementation of other components can rely on the behavior of any RL components while still allowing the RL pipelines to produce better-trained agents. Second, users expect pipelines to behave in generally the same way over time, even after models are updated. For example, if a non-robust autonomous driving agent is retrained with slightly updated data and behaves differently, user expectations are subverted.

A third reason is that robust pipelines are useful in *counterfactual explanation* (CE), an explanation method for XAI in general and XRL specifically Byrne (2019); Madumal et al. (2020); Elizalde et al. (2008); Karimi et al. (2020); Chou et al. (2022); van der Waa et al. (2018). CE explains agent actions by reasoning about how interventions on state affect which actions are selected by that agent. If an RL pipeline generates a variety of outputs in a single intervened state, then explaining the effect of that intervention is difficult. For example, a CE system may infer that a stop sign causes autonomous vehicles to stop at an intersection. For a robust pipeline, this may be true. However, a non-robust pipeline may produce agents that stop, continue forward, or turn right, making the CE invalid. Existing CE methods do not explicitly state the need for robustness, but it is implicitly assumed that counterfactual explanations are reproducible for a generic RL pipeline. By explicitly measuring the OR of a pipeline, we provide a means to validate these assumptions.

Why focus on *offline robustness*, rather than on-policy robustness? Measuring robustness on environments that are not necessarily in-distribution is important for users who want to deploy agents in these scenarios and are interested in the benefits of robustness discussed above. CE users specifically may want to verify the effect of an intervention to see if the CE system is working properly. Out-of-distribution states allow users to examine agent behavior and can help users identify if an agent has memorized the training environment by seeing if it performs poorly on an out-of-distribution environment. Observing agents before and after intervention or in- and out-of-distribution can help to differentiate between memorized and strategic behavior (Witty et al., 2021; Cobbe et al., 2019; Packer et al., 2018; Littman & Szepesvári, 1996). Some policies from non-robust pipelines may appear to be strategic, while others may appear memorized.

We find that most policies of high-performing RL agents playing Atari games are not produced by robust pipelines. To evaluate this claim and differentiate these policies from others, we propose a quantitative measure of OR based on the dissimilarity of policies' actions for both familiar and unfamiliar states. Intuitively, the OR of an RL training pipeline is a measure of how *rarely* agent actions vary on an intervened state across policies generated by independent pipeline instances. If policies produced by multiple pipeline instances tend to take the same action in a given (intervened) state, then the training pipeline is robust. As an example, consider three Atari agents *Agent A, Agent B, and Agent C* trained using different instances of the same pipeline. OR measures the dissimilarity of the actions taken by Agents A, B and C. If the actions are similar, then the RL training pipeline has high OR and may increase the trustworthiness of explanations applied to these policies and other policies from this pipeline.

Specifically, we:

1. Provide a formal definition of offline robustness and a specific measurement of OR;

2. Analyze how OR varies by training algorithm, training environment, and amount of training; and

3. Demonstrate that OR does not increase monotonically with amount of training and is not necessarily correlated with on-policy performance.

The rest of paper is organized as follows: in Section 2, we introduce related work. We describe OR and how to measure it in Section 3. Our experiments and results are explained in detail in Section 4 and 5. Finally, we discuss implications of this work for XRL and future directions in Sections 5 and 6.

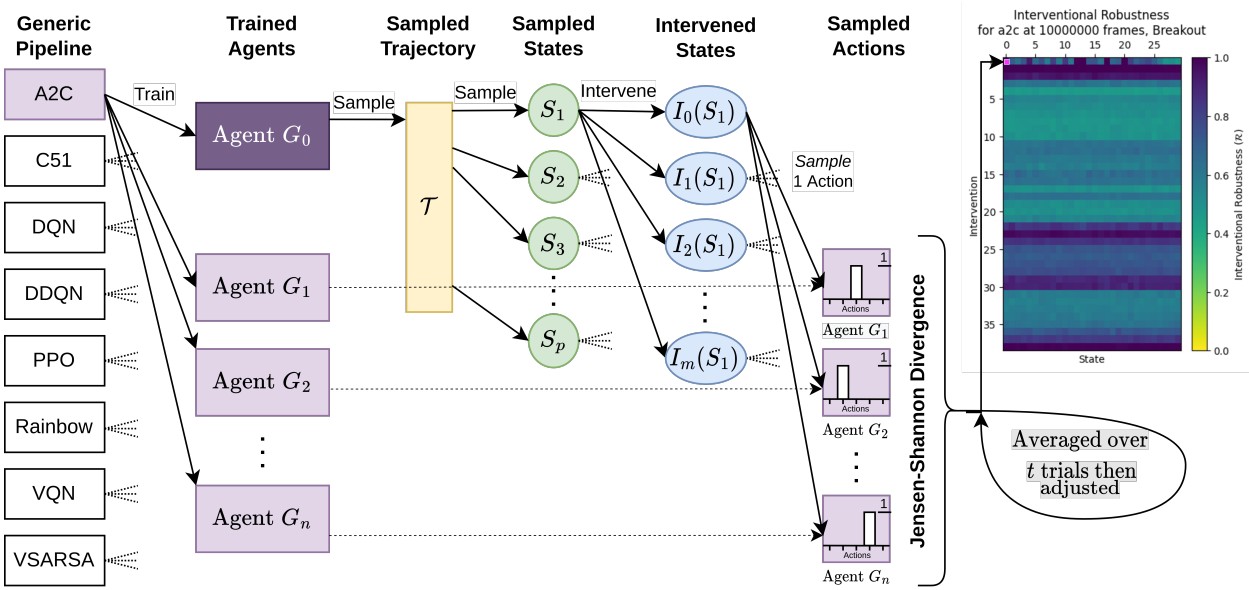

Figure 2: **Offline Robustness Measure.** This graphic describes the method from Section 3 evaluating OR on different RL pipelines for a given environment. Eleven agents ($\mathbb{G}$) of each learning algorithm are trained on the environment of interest, and a trajectory is sampled from one, $G_0$. From this trajectory, $p = 30$ states are sampled uniformly randomly. The adjusted JS Divergence is computed for each intervened state and the original state with no intervention using the distribution over actions sampled from the pool of trained agents as described in Section 3. For non-greedy, stochastic policies, this sampling is performed $t = 30$ times and averaged to get an accurate estimate of $\mathcal{R}$, while greedy policies require only one sample. In the final plotted output, each column represents a unique state sampled from the trajectory and each row corresponds to that state under a distinct intervention. The top-most row is the null intervention. A full description of this process can be found in Algorithms 3 and 2.

## 2 Background and Related Work

A number of research papers in reinforcement learning acknowledge the differences in the policies produced by an RL pipeline, though they use a variety of terminology. While some authors refer to the *consistency* of RL pipelines, others define and measure *stability*, *variability*, *reliability* and *robustness*. In most cases, variability refers to differences in the performance or actions of a stochastic policy (Henderson et al., 2018; Chan et al., 2020). Stability refers to variability within training runs, and is often used from a control theoretic perspective to signify the ability of a trained agent to safely recover from exploratory actions (Buşoniu et al., 2018; Berkenkamp et al., 2017; Jin & Lavaei, 2020). Robustness generally refers to the ability of an RL agent to generalize in out-of-distribution scenarios, maintaining its performance when there are plausible perturbations to the environment Al-Nima et al. (2021); Marc Fischer & Vechev (2019); Ma et al. (2018). A majority of the research in these areas tends to focus on the performance of the RL agent rather than the actions of the policy resulting from a pipeline instance. While similarities in the performance of the agent are beneficial from the perspective of deployment, there is little focus on actions taken by the agents that lead to a specific performance. Our work is different from the existing literature in two important ways: we measure the dissimilarities in the actions of RL agents *under intervention*, and we measure robustness of a distribution of policies that are generated from a *generic* RL pipeline as opposed to the results of a pipeline instance (a single policy).

Prior work in XAI and XRL refers to the impact of non-robustness on the trustworthiness of the pipeline as a whole. Typically, the dissimilarities in behavior stem from multiple sources of variability: hyperparameters in the training procedure, random seeds, environmental characteristics, network architectures, and underspecification in the pipeline (Henderson et al., 2018; Clary et al., 2018; Islam et al., 2017; D'Amour

et al., 2020). *Underspecification* occurs when a pipeline returns policies that have different strategies but perform equivalently well. Offline robustness of this pipeline would be low, even though the performance of the pipeline's agents are similar. This effect has also been referenced as the "Rashomon effect" or the multiplicity of good models outside of RL (Breiman, 2001; Rudin et al., 2022). The dissimilarities among pipeline instances manifest in different ways, such as unpredictable or noisy fluctuations of performance during training, irreproducible performances across training runs, and variability in performance across rollouts of a policy (Chan et al., 2020). In our case, they manifest as dissimilarities in the actions of RL agents that are a result of generating policies from multiple instances of the same pipeline, as well as dissimilarities in the actions of agents under slight changes to the environment (interventions).

There has been some work in exploring the connection between changes in the policy (termed *policy churn*) throughout training for a single pipeline instance (Schaul et al., 2022). This work highlights a significant amount of churn for some deep RL agents despite the policy reaching its optimal value. These changes in the policy occur despite little to no increase in the performance of the agent. While their measure explores changes in the policy for a single pipeline instance, ours is the first to provide an explicit measure of policy differences for a generic RL pipeline.

## 3 Offline Robustness

Broadly, OR measures the similarity of actions of agents acting under policies that are produced by multiple instances of an RL pipeline. We measure OR to detect how incidental differences in the pipeline affect the agent actions under intervention.

### 3.1 Example in Atari

Consider an example where an RL pipeline is used to train agents that play the Atari game *Space Invaders*. The only differences among the agents produced by the generic pipeline are incidental features of the training procedure, determined by the random seed. The goal of Space Invaders is for an agent to shoot incoming enemies before they reach the ground, while avoiding enemy shots by dodging the shots or hiding under a shelter. An agent can move left, right, stay in place, and shoot. To measure OR for the RL pipeline, we observe how similarly agents act when we intervene in the Space Invaders environment. Interventions are direct, irreversible changes made to the state of a game by assigning values to one or several variables in it. An example intervention in Space Invaders is the removal of a shelter from the screen, reducing the agent's ability to hide from enemy shots. The more similarly the agents behave when we make a specific intervention, the higher OR is with respect to that intervention; the less similarly they behave, the lower OR is with respect to that intervention.

### 3.2 Description of Offline Robustness

For our purposes, we define interventions in the context of reinforcement learning as follows:

**Definition 3.1** (Intervention). An intervention is a change to one or several variable values of an environment at a single timestep. These are applied in such a way that future states maintain these changes and that the response of agents can be measured following the application. Interventions are often, but not always, outside of the training distribution.

We define offline robustness (OR) as a property of an RL pipeline:

**Definition 3.2** (Offline Robustness). An RL pipeline is robust to the degree that it produces a group of agents $\mathbb{G}$ that choose actions according to the same action distribution from an offline test environment in $I$ on a select state in $S$, as opposed to an action distribution that varies from agent to agent across the $S \times I$ offline test states.

In this paper, the offline environments $I$ are generated by intervening on the original environment to get environments that are largely out-of-distribution. For example, a group of 10 DQN agents trained on different random seeds may behave very similarly given a particular intervention, while a group of 10 VSARSA agents

may behave very differently from one another given that same intervention. The DQN agents would be considered to be highly robust, while the VSARSA agents would not. Note that OR is distinct from performance. A high-performing group of 10 agents from an underspecified pipeline could have 10 different high-performing strategies. This would almost certainly result in low OR despite their uniformly high performance. From an explainability perspective, variance in action selection limits the ability of counterfactual-explanation methods to provide useful explanations, as well as reducing confidence that those explanations will apply to other agents produced by that pipeline (e.g. after retraining or online learning in deployment).

More examples of the way performance and OR are related can be found in Table 1. High performing but non-robust pipelines may be difficult to explain but useful in practice, whereas highly robust pipelines of varied performance can yield explanations of an agent that are more applicable to other agents produced by that pipeline. We include the best examples of each from Figure 3 to illustrate this relationship. Note that there is not necessarily a trade-off between performance and OR. It is possible for an agent to be both high-performing and robust or low-performing and non-robust, as shown in Table 1. For example, agents that take uniformly random actions would have both low performance and low OR.

We measure OR formally as follows:

**Definition 3.3** (Robustness Measure ($\mathcal{R}$)). Given a set of $n$ independently trained agents $\mathbb{G} = \{G_1, ..., G_n\}$ from the same generic RL pipeline and an offline state of interest $s \in S \times I$, $\mathcal{R}(\mathbb{G}, s) \in [0, 1]$ measures how similarly the agents $\mathbb{G}$ act in state $s$ where a value closer to 1 indicates higher similarity, and therefore higher robustness. Similarity is measured by comparing the action distributions of each agent in a particular state $s$ as described in Algorithm 1.

To compare these distributions, we implement $\mathcal{R}$ using Jensen-Shannon Divergence (JSD), although other measures of divergence could be used (Lin, 1991). We use JSD because it is symmetric; that is, the value is the same when comparing distribution $a$ to distribution $b$ as when comparing $b$ to $a$, and can compare multiple distributions at once (i.e. one from each agent). JSD also has finite bounds, so we can normalize and compare the values from different RL pipelines directly. For these reasons, the use of Jensen-Shannon Divergence is an implementation choice that is well-suited for the task.

The approach for calculating $\mathcal{R}$ using a group of agents produced by the same RL pipeline is laid out in Algorithms 1, 2, and 3. To apply the measure, we train multiple agents from a generic pipeline and sample their actions in specific states by applying interventions of interest. Intuitively, Algorithm 2 computes a value of $\mathcal{R}$ for each state-intervention pair by sampling a distribution of actions from each agent and computing an adjusted JSD value from those distributions (see Algorithm 1 for details).

---

**Algorithm 1:** Calculating $\mathcal{R}$ value for a given state $s$ (intervened) and a set of agents $\mathbb{G}$.

**Input:** $s$: Intervened state, $\mathbb{G}$: Set of agents,
$\quad\quad\quad$ $t$: the number of samples to use for a stochastic policy, for deterministic policy $t = 1$
**Output:** $\mathcal{R}$: Offline robustness value for $\mathbb{G}$ in $s$
**begin**
$\quad$ $\mathbf{r} \leftarrow \text{ARRAY}(t)$ $\quad\quad\quad\quad\quad\quad\quad\quad\quad\quad\quad\quad\quad\quad\quad\quad\quad\quad\quad\quad$ `// List of t JSD samples`
$\quad$ **for** $i \in \{1, \ldots, t\}$ **do** $\quad\quad\quad\quad\quad\quad\quad\quad\quad\quad\quad\quad\quad\quad\quad\quad\quad\quad$ `// samples`
$\quad\quad$ $\mathbf{a} \leftarrow \text{ARRAY}(n)$ $\quad\quad\quad\quad\quad\quad\quad\quad\quad\quad\quad\quad\quad\quad$ `// List of actions from n agents`
$\quad\quad$ **for** $k \in \{1, \ldots, n\}$ **do** $\quad\quad\quad\quad\quad\quad\quad\quad\quad\quad\quad\quad\quad\quad\quad$ `// agents`
$\quad\quad\quad$ $\mathbf{a}_k \sim G_k(s)$ $\quad\quad\quad\quad\quad\quad$ `// Sample action from policy for agent` $G_k$ `in state` $s$
$\quad\quad$ **end**
$\quad\quad$ $d \leftarrow \text{JSDIVERGENCE}(\mathbf{a})$
$\quad\quad$ $d \leftarrow d/log_2(n)$ $\quad\quad\quad\quad\quad\quad\quad\quad\quad\quad\quad\quad\quad\quad$ `// Normalized between` $[0, 1]$
$\quad\quad$ $\mathbf{r}_i \leftarrow 1 - d$ $\quad\quad\quad\quad\quad\quad\quad\quad\quad\quad\quad\quad\quad\quad\quad$ `// Robustness is similarity`
$\quad$ **end**
$\quad$ **return** $\text{MEAN}(\mathbf{r})$
**end**

---

Table 1: The interaction between performance and OR value $\mathcal{R}$

| | | Offline Robustness | |
|---|---|---|---|
| | | **Low** | **High** |
| **Performance** | **Low** | Agents are Incompetent and Diverse (e.g., PPO @ 1e7 frames) | Agents are Incompetent and Similar (e.g., VQN @ 1e7 frames) |
| | **High** | Agents are Competent and Diverse (e.g., DQN @ 1e7 frames) | Agents are Competent and Similar (e.g., Rainbow @ 1e7 frames) |

Our measure supports both deterministic and stochastic policies. For agents that select actions greedily, where the same action will always be chosen for a given state, the distribution is fixed with all of the probability on one action. We collect these deterministic distributions from each agent and use the divergence measure to compare them and compute the $\mathcal{R}$ value. To support stochastic policies that sample randomly across the actions according to a distribution informed by the policy applied to the state of interest, we average this computation over several trials to account for the variation of the policy.

---

**Algorithm 2:** The procedure used to estimate $\mathcal{R}$ for set of agents $\mathbb{G}$ on an environment $\mathcal{E}$. This polynomial time algorithm produces a matrix of values $\boldsymbol{R}$, as illustrated in Figure 2.

---

**Input:** $\mathbb{G} = \{G_1, \ldots, G_n\}$: a set of $n$ trained agents
$\quad\quad$ $I = \{I_0, I_1, \ldots, I_m\}$ : an array of $m + 1$ interventions, including the null intervention $I_0$
$\quad\quad$ $S = \{S_1, \ldots, S_p\}$: the array of $p$ states from Algorithm 3
$\quad\quad$ $t$: the number of samples to use for a stochastic policy, for deterministic policy $t = 1$
**Output:** $\boldsymbol{R} = \mathcal{R}(\mathbb{G}) \in [0,1]^{p \times m}$ $\quad\quad\quad\quad\quad\quad\quad\quad\quad\quad$ `// OR measure`
**begin**
$\quad$ **for** $k \in \{1, \ldots, p\}$ **do** $\quad\quad\quad\quad\quad\quad\quad\quad\quad\quad\quad\quad\quad\quad\quad$ `// states`
$\quad\quad$ **for** $j \in \{0, 1, 2, \ldots, m\}$ **do** $\quad\quad\quad\quad\quad\quad\quad\quad\quad\quad\quad$ `// interventions`
$\quad\quad\quad$ $s \leftarrow I_j(S_k)$ $\quad\quad\quad\quad\quad\quad\quad\quad\quad\quad$ `// apply intervention to state`
$\quad\quad\quad$ $\boldsymbol{R}_{k,j} \leftarrow \text{ALGORITHM1}(s, \mathbb{G}, t)$
$\quad\quad$ **end**
$\quad$ **end**
$\quad$ **return** $\boldsymbol{R}$
**end**

---

When using OR in practice, offline data can be selected in several ways. System designers applying CE will always have a specific set of interventions, which the system will employ to explain agent actions. (In CE, interventions correspond directly to the components of explanations provided to users). If an extremely large number of interventions are defined by a given CE system, they can be sampled randomly or chosen based on properties of the environment.

System designers not applying CE will still need to produce $p$ offline states. The states chosen would depend on the designer's goals. For example, a designer may want to collect a dataset of states that may be encountered at test time and measure the OR on that dataset.

Although this paper calculates OR for the states in the original training environment, as well as each intervened environment, system designers may want to use OR in a variety of ways. For example, a designer may want to use the average OR on their offline dataset, as shown in the tables in Appendix E. If high

robustness is critical to their application, a designer might want to measure the minimum OR for a state in their offline dataset.

## 3.3 State Sampling

OR is measured for a specified set of offline states. System designers may have a set of states of interest, but in the absence of domain knowledge, states sampled from an agent with an effective policy will suffice for sampling likely states. Further, there are often too many possible states in an environment to calculate the OR for each, so in Algorithm 3 we approximate the full set of states by sampling a set of states from a trajectory produced by one additional agent. Sampling from a trajectory made by an agent from the same RL pipeline for which we want to measure OR produces a set of states likely to be encountered by agents from that pipeline, thus ensuring the plausibility of these training states.

---

**Algorithm 3:** The procedure used to sample the $p$ states from an agent not used in Algorithm 2.

---

**Input:** $\mathcal{E}$: environment of interest
$\quad\quad$ $G$: a trained agent
$\quad\quad$ $p$: The number of states to sample
**Output:** $S$: an array of $p$ sampled states
**begin**
$\quad$ $\mathcal{T} \leftarrow \mathcal{E}(G)$ $\quad\quad\quad\quad\quad\quad\quad\quad\quad\quad\quad$ // Collect the states from $G$ interacting in $\mathcal{E}$
$\quad$ $S \leftarrow \text{ARRAY}(p)$
$\quad$ **for** $k \in \{1, \ldots, p\}$ **do** $\quad\quad\quad\quad\quad\quad\quad\quad\quad\quad\quad$ // states
$\quad\quad$ | $\quad$ $S_k \leftarrow \mathcal{T}[\text{UNIFORMRANDOMINTEGER}(0, |\mathcal{T}|)]$
$\quad$ **end**
$\quad$ **return** $S$
**end**

---

## 4 Experiments

We performed experiments to explore whether several commonly used deep RL pipelines are robust—that is, whether pipelines produce policies (and therefore agents) that act similarly on offline data of interest, despite incidental differences in the training procedure. In our experiments, we use the Autonomous Learning Library (ALL) (Nota, 2020) to train RL agents on intervenable Atari environments provided by the Toybox library (Foley et al., 2018). We then evaluate the OR of those agents using Algorithm 2. A schematic of our experimental method is shown in Figure 2. We estimate the $\mathcal{R}$ values for eight RL pipelines producing $n = 10$ agents for $p = 30$ states in three Atari environments (Space Invaders, Amidar, and Breakout). The number of interventions $m$ varies per environment. All RL training pipelines used the default hyperparameters given by ALL.

For each environment, we defined dozens of interventions on state that could affect the agents' experience and actions while remaining reasonably similar to the actual game, such that a human could easily continue to play under the intervention. When defining interventions, we attempted to avoid fundamental changes to the "physics" of the game, and instead changed less fundamental aspects such as the positions of existing objects. Note that the intervenable Atari environment allows for underlying state changes to the simulator, not just pixel-level interventions. Our code is open source and can be found on GitHub.[1]

### 4.1 Space Invaders

In Space Invaders, an agent moves left and right and hides behind shields to avoid enemy lasers. Some interventions (see Appendix, Table 4) remove one or more enemies, which changes where and how often shots are fired but also limits the maximum agent score. Some changes are superficial, such as changing game icons (intervention 87). Other interventions check for memorization, such as moving the locations of

---

[1]GitHub link redacted

shields to check whether an agent has learned to hide under shields generally rather than merely moving to certain pixel coordinates (interventions 36-45).

Results of these experiments are shown in Figure 3. The results exhibit several interesting patterns when comparing training time and $\mathcal{R}$ values:

- *Inexperienced agents often have high robustness by default*: Early in the training of C51, DQN, and DDQN, these pipelines have high OR and poor performance, indicating that these agents often choose actions similarly given only small amounts of training data.

- *Highly trained agents often behave similarly, but still perform relatively poorly*: Late in the training of VQN and VSARSA, these pipelines have high OR as well, meaning the policies converge to similar action distributions after much training. However, relative to other pipelines, they have low performance.

- *Some pipelines produce relatively performant agents that are robust*: Rainbow, an improved version of C51 and other DQN-based methods, converges to a relatively high performance and relatively high OR.

A natural question to ask once the $\mathcal{R}$ values are computed for each state and set of agents is how robustness *changes* on the offline data. To answer this question, we compared each intervened state's $\mathcal{R}$ value to its pre-intervention state's value. To demonstrate the relative difference we normalized the values of Figure 3, where the top row of each plot is the unintervened state, normalized to 0. Each other cell in the plot is then between $[-1, 1]$, where a positive score indicates that the pipeline produces agents that are more robust under intervention for this (state, intervention) pair. In other words, robustness was higher for the intervened state than the original. A negative score then indicates that the intervention decreases robustness among the agents produced by the pipeline. The results of this comparison are illustrated in Figure 4.

### 4.2   Amidar

Amidar is similar to Pac-Man, where an avatar avoids enemies along a set corridors. To clear the level, the avatar must traverse all segments of the corridors. Adding and removing corridor segments changes the maximum possible score. The interventions again check for memorization. If the agent acts uniform randomly when starting in a different location, it has likely memorized information about its initial position and has not generalized. Similarly, if an agent becomes stuck when segment of corridor is removed, it has not learned navigation and has merely memorized the board. The plots of $\mathcal{R}$ values can be found in Appendix C.1, Figure 8. These results are very similar to those of Space Invaders, although Rainbow provides the best performance and has a moderate, rather than high, OR. DQN and DDQN are also more robust than in Space Invaders.

### 4.3   Breakout

In Breakout, the player operates a paddle and uses a ball to break blocks. If all blocks are broken, the player wins, and if the ball is lost, the player loses. Dropping rows and columns of blocks changes the maximum possible score. As in previous environments, the interventions check for memorization. The plots of $\mathcal{R}$ values can be found in Appendix C.2, Figure 9. Overall, Breakout added supporting evidence for the trends observed in the other two environments.

## 5   Discussion

Overall, RL pipelines using the same underlying algorithm have similar $\mathcal{R}$ values across environments, and $\mathcal{R}$ often increases with more training. $\mathcal{R}$ values are recorded in Appendix E. In the environments tested, VSARSA and VQN in any environment and Rainbow in Space Invaders are particularly robust, though VSARSA and VQN have consistently low performance. Typically, however, the highest performing agents are less robust than middle-of-the-pack agents, as shown in Figures 3, 8, and 9.

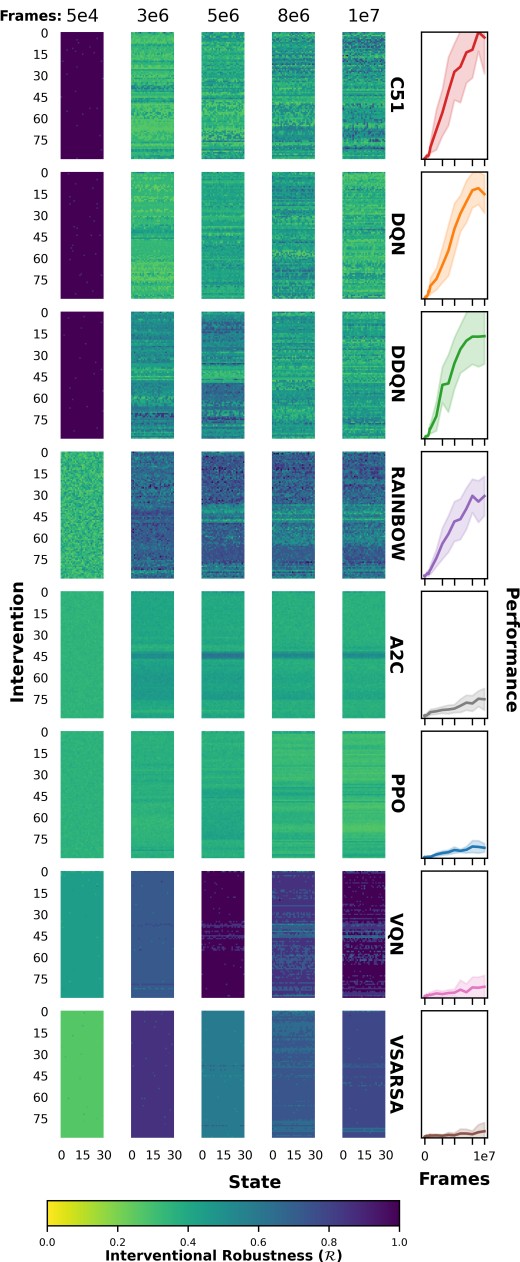

Figure 3: **Offline Robustness: Space Invaders** Table of $\mathcal{R}$ values for agents from different pipelines at different checkpoints of training. The plots on the right edge correspond to the score as the group of agents from the pipeline were trained, and each of those plots has the same y-axis. The rows of plots are sorted according to performance, top being highest performing, bottom being lowest. The top three performers C51, DQN, and DDQN demonstrate high OR when relatively untrained. Rainbow is the highest performing agent that is also relatively robust. VQN and VSARSA demonstrate consistent OR after some training, but their performance is the lowest, as illustrated by the performance plots in the rightmost column.

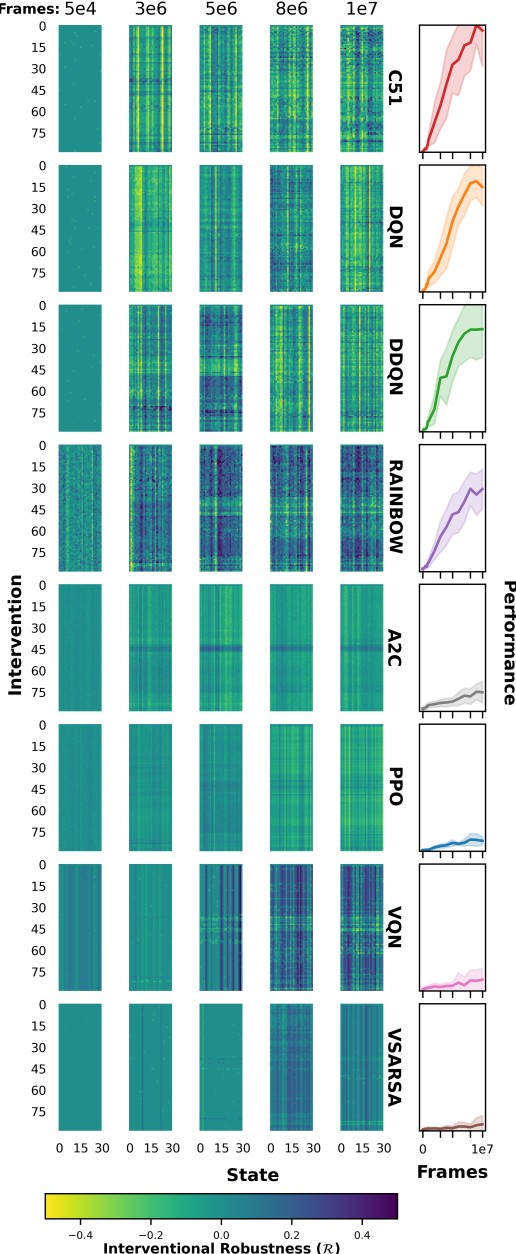

Figure 4: **Relative Offline Robustness: Space Invaders** Table of relative $\mathcal{R}$ values compared to the states prior to intervention for different pipelines at different checkpoints in training. As in Figure 3, the rightmost column of plots show the performance of the agents from each pipeline, the columns left to right increase the level of training that the agents received, and each row of plots corresponds to a different pipeline. In these plots, the most noteworthy details are that under intervention, the pipelines that appear more robust under intervention on average are Rainbow, VSARSA, and VQN. The rest average negative relative $\mathcal{R}$ values, so they are less robust under intervention. The range is truncated at -0.5 and 0.5 to highlight the differences between the pipelines. The proportion of points outside the bounds for this plot are 0.157%, meaning very little information is lost by limiting the range. The numerical values corresponding to these experiments can be found in Table 7.

More specifically, VQN and VSARSA have similar $\mathcal{R}$ values in Amidar and Space Invaders, though they diverge somewhat in Breakout. DQN-based methods, including DQN, DDQN, and C51, have similar $\mathcal{R}$ values in all three environments. The only exception is the DQN-based method Rainbow, which has an OR that starts low and increases over time. Other DQN methods start with high OR, which quickly drops and then recovers slowly.

Further inspection reveals that pipelines that are robust early in training are simply choosing the same default action over and over until more training information becomes available. As the agents become better at the game and score higher, they start choosing different actions, and their OR drops. Slowly, the agents often converge to more similar strategies, and the OR improves again. For example, the agents in Space Invaders may learn to use shields more effectively, so they may be more likely to move left in a given state to duck behind a shield. It is also possible that some pipelines are underspecified, so agents could learn different high-scoring strategies. In an RL setting, underspecification lowers OR since agents may take different actions in the same states.

The plots also reveal differences in values of OR corresponding to different interventions. For example, C51 in Space Invaders has a higher OR for some interventions compared to others, as indicated by the dark horizontal bands in Figure 3. When these interventions are used in CE, we can have correspondingly higher confidence in their ability to support generalized reasoning of users.

In addition to the types of plot shown in Figure 3, we produced results that normalized each $\mathcal{R}$ value by the $\mathcal{R}$ value of the unintervened state (see Appendix D, Figures 11 to 13). Those results indicate that the OR for some states generally increased when intervened on, while the OR of others decreased. How the agents reacted to intervention depended heavily on the state and algorithm.

Differences in the value of $\mathcal{R}$ for different pipelines, as well as how $\mathcal{R}$ changes over the course of training in these pipelines, indicates the utility of measuring OR. Robustness is a useful property for explainable and reliable pipelines. However, high performance does not indicate high OR and vice versa, so measuring OR itself is useful for comparing pipelines. Users can use $\mathcal{R}$ values to decide whether they should delegate to or deploy a pipeline, which pieces of software could safely rely on a pipeline's outputs, how often to run explanation methods, and whether they should run CE specifically.

## 6 Conclusions and Future Work

We define and study offline robustness (OR), a measure of how sensitive the agents produced by RL training pipelines are to incidental differences in the training procedure. We demonstrate that OR varies substantially based on the environment, training algorithm, amount of training, and type of offline data (i.e. intervention). Somewhat surprisingly, we demonstrate that agent performance is not strongly predictive of OR. High-performing agents can have low OR and vice versa.

Along with performance, OR can be used as a tool for choosing pipelines for delegation or deployment. For example, a system designer may choose a pipeline with reasonable performance and high OR if they want a very explainable system. Pipelines with high OR have a variety of useful properties. They behave similarly over time, fulfilling users' expectations, and other software systems can rely on their outputs. Robust pipelines are good candidates for CE, and their previous explanations remain relevant over time. We provide a quantitative measure of OR that can be used to select pipelines with high OR that also produce agents with high performance. In addition, this work provides an important foundation for future studies of new robustness measures and ways to increase robustness.

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

# A    Interventions

## A.1    Amidar

Table 2: Description of the 70 interventions $I$ used in the experiments with Amidar.

| Number | Intervention |
| --- | --- |
| 0-31 | Remove one tile from the board |
| 32-56 | Add one tile to the board |
| 57-61 | Drop one enemy |
| 62-65 | Start an enemy elsewhere on the board |
| 66-69 | Start the agent elsewhere on the board |

## A.2    Breakout

Table 3: Description of the 39 interventions $I$ used in the experiments with Breakout.

| Number | Intervention |
| --- | --- |
| 0-4 | Change the paddle width |
| 5-10 | Change the paddle speed |
| 11-14 | Change the paddle start position |
| 15-20 | Drop a row of bricks |
| 21-39 | Drop a column of bricks |
| 25-38 | Drop a column of bricks |

### A.3 Space Invaders

Table 4: Description of the 88 interventions $I$ used in the experiments with Space Invaders.

| Number | Intervention |
|--------|--------------|
| 0-35 | Drop one enemy from each location independently |
| 36-45 | Shift protective shields uniformly |
| 46-74 | Shift where agent starts on x-axis of game |
| 75-86 | Drop entire row or column of enemies at a time. |
| 87-87 | Flip shield icons vertically to change pixel inputs, but maintain level of defense provided. |

# B  Agent Performance

## B.1  Amidar

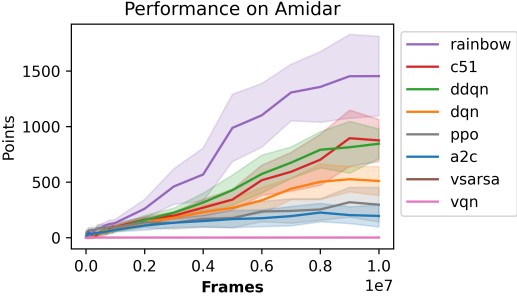

Figure 5: Performance of agents from each pipeline on Amidar, sorted in legend from highest to lowest performing.

## B.2  Breakout

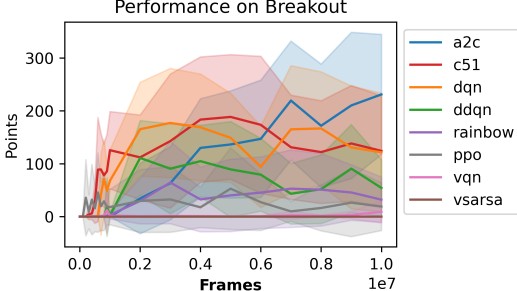

Figure 6: Performance of agent from each pipeline on Breakout, sorted in legend from highest to lowest performing.

### B.3 Space Invaders

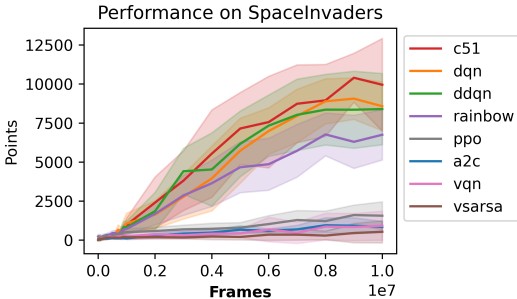

Figure 7: Performance of agent from each pipeline on Space Invaders, sorted in legend from highest to lowest performing.

## C    Offline Robustness Plots

### C.1    Amidar

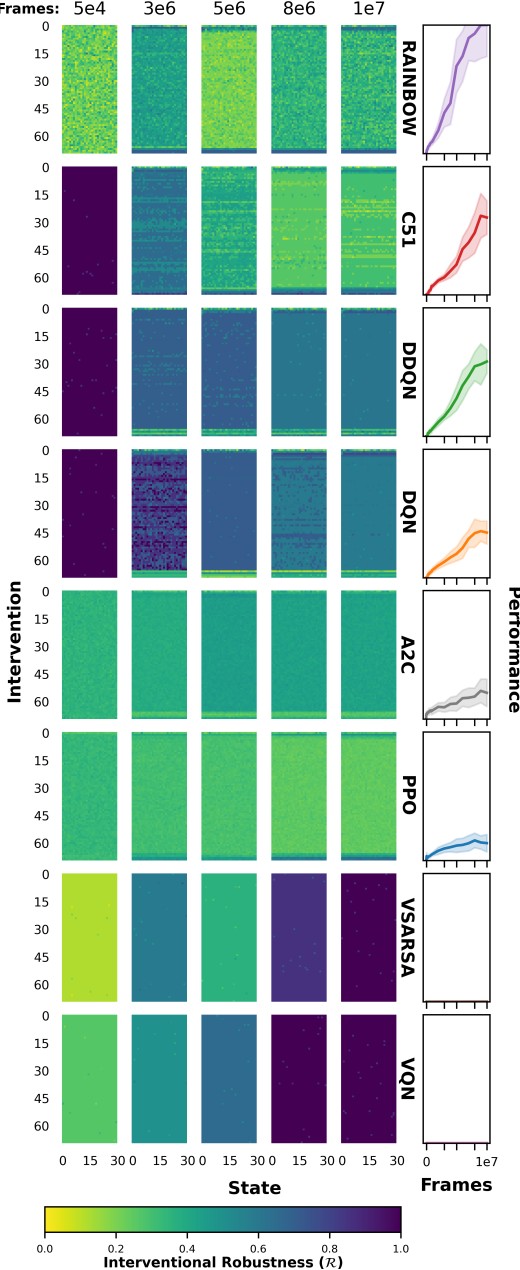

Figure 8: Table of $\mathcal{R}$ values for agents from each pipeline at different checkpoints of training, plotted as a color image matrix.

## C.2 Breakout

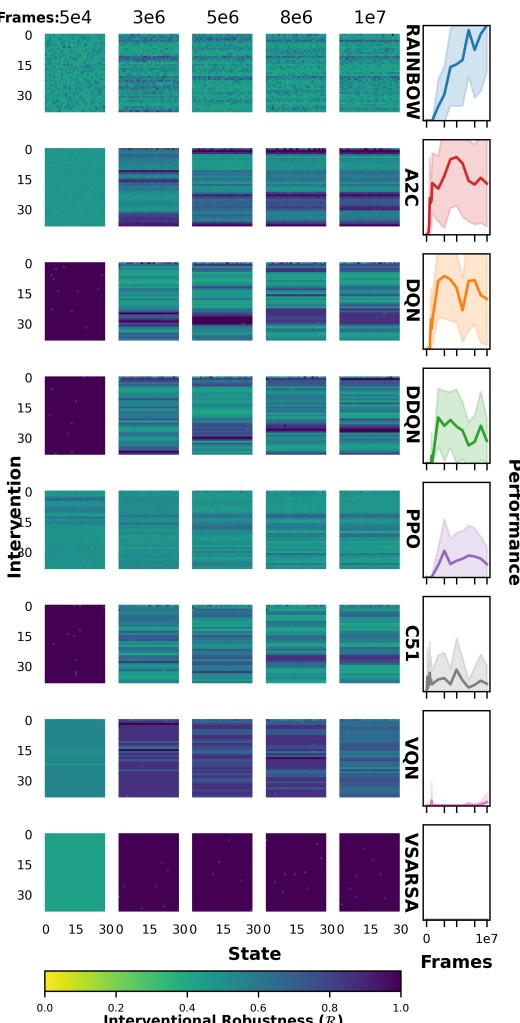

Figure 9: Table of $\mathcal{R}$ values for agents from each pipeline at different checkpoints of training, plotted as a color image matrix.

## C.3 Space Invaders

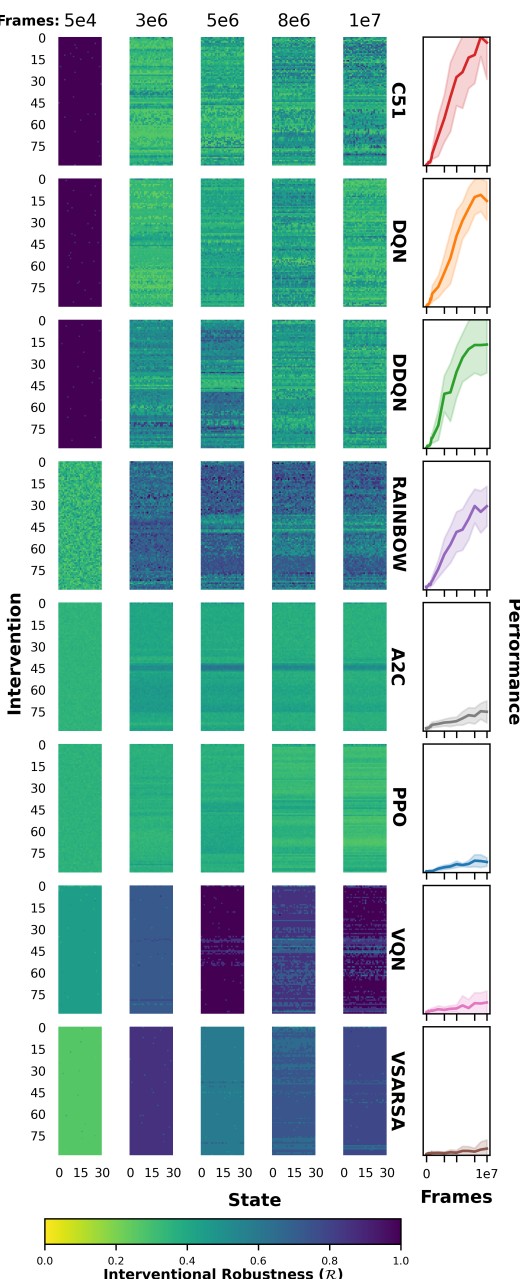

Figure 10: Table of $\mathcal{R}$ values for agents from each pipeline at different checkpoints of training, plotted as a color image matrix.

# D    Normalized Offline Robustness Plots

The following figures contain tables of normalized $\mathcal{R}$ values for each pipeline at several training checkpoints, shown as color images. They are normalized with respect to the unintervened state $I_0(s)$, the first row. Unlike the unnormalized plots, the scale is from -1 to 1. Zero means that the intervened state is as robust at the original state, while 1 means it is more robust than the original, and -1 means it is less robust than the original.

## D.1 Amidar

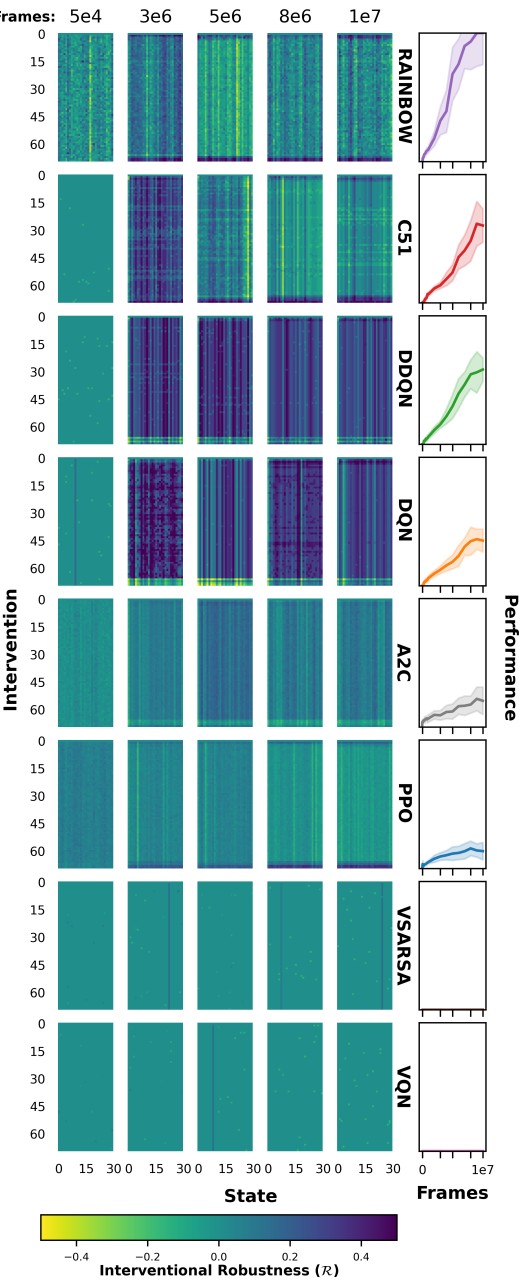

Figure 11: Table of normalized $\mathcal{R}$ values for agents from each pipeline at different checkpoints of training, plotted as a color image matrix. Each image is normalized column-wise with respect to the unintervened state $I_0(s)$, the first row. The range is truncated at -0.5 and 0.5 to highlight the differences among the pipelines. The proportion of points outside the bounds for this plot are 1.243%, meaning very little information is lost by limiting the range.

**D.2  Breakout**

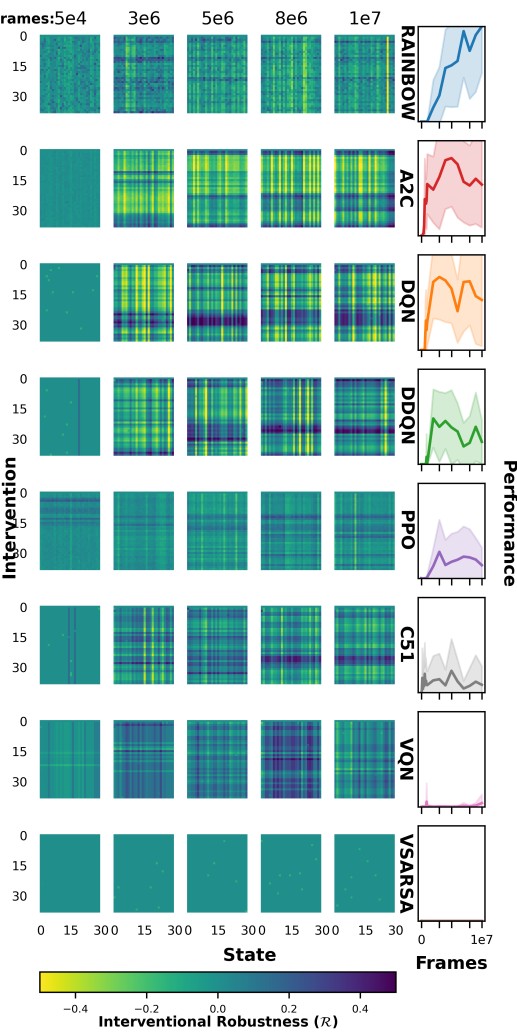

Figure 12: Table of normalized $\mathcal{R}$ values for agents from each pipeline at different checkpoints of training, plotted as a color image matrix. Each image is normalized column-wise with respect to the unintervened state $I_0(s)$, the first row. The range is truncated at -0.5 and 0.5 to highlight the differences among the pipelines. The proportion of points outside the bounds for this plot are 0.8098%, meaning very little information is lost by limiting the range.

### D.3 Space Invaders

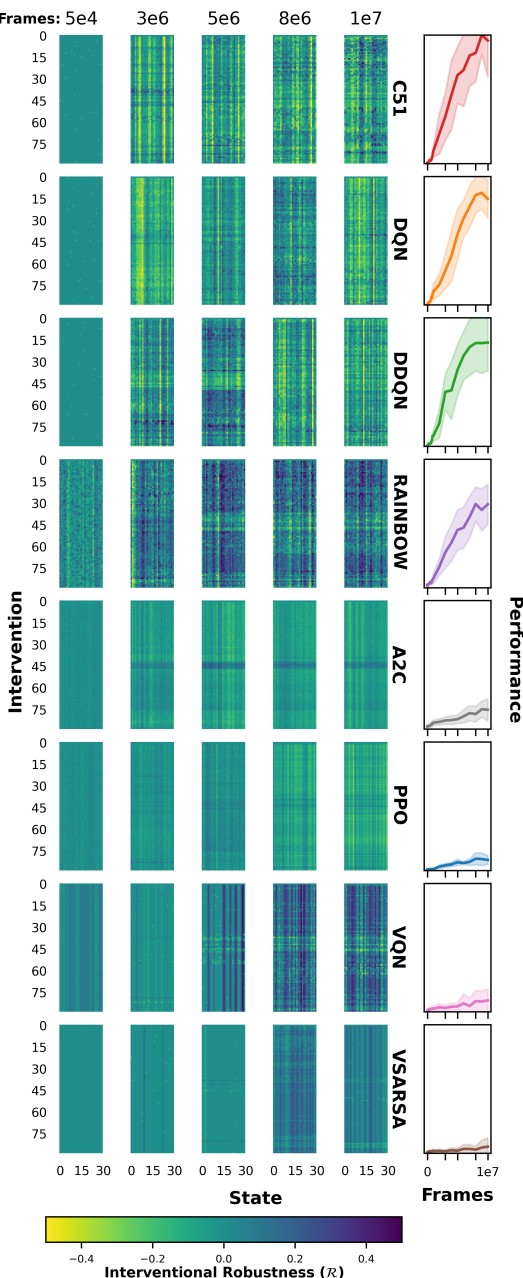

Figure 13: Table of normalized $\mathcal{R}$ values for each pipeline from different checkpoints of training, plotted as a color image matrix. Each image is normalized column-wise with respect to the unintervened state $I_0(s)$, the first row.

# E   Offline Robustness ($\mathcal{R}$) Values

The values in the tables below are the average $\mathcal{R}$ values for each image above. As above, the values in the "Normalized" column are the "Intervened" column, normalized with the unintervened state. They were normalized for each state-intervention pair prior to averaging.

## E.1   Amidar

Table 5: Table of $\mathcal{R}$ values for Amidar.

| Algorithm | Frames | Original | Intervened | Normalized |
|---|---|---|---|---|
| a2c | 5e4 | 0.347 | 0.352 | 0.005 |
| a2c | 3e6 | 0.313 | 0.377 | 0.065 |
| a2c | 5e6 | 0.300 | 0.434 | 0.134 |
| a2c | 8e6 | 0.349 | 0.421 | 0.072 |
| a2c | 1e7 | 0.335 | 0.414 | 0.080 |
| dqn | 5e4 | 0.995 | 0.999 | 0.003 |
| dqn | 3e6 | 0.446 | 0.791 | 0.345 |
| dqn | 5e6 | 0.452 | 0.680 | 0.228 |
| dqn | 8e6 | 0.301 | 0.604 | 0.304 |
| dqn | 1e7 | 0.332 | 0.582 | 0.251 |
| ddqn | 5e4 | 1.000 | 0.998 | -0.002 |
| ddqn | 3e6 | 0.394 | 0.673 | 0.279 |
| ddqn | 5e6 | 0.366 | 0.679 | 0.313 |
| ddqn | 8e6 | 0.328 | 0.598 | 0.271 |
| ddqn | 1e7 | 0.349 | 0.600 | 0.251 |
| c51 | 5e4 | 1.000 | 0.999 | -0.001 |
| c51 | 3e6 | 0.341 | 0.613 | 0.272 |
| c51 | 5e6 | 0.321 | 0.390 | 0.069 |
| c51 | 8e6 | 0.290 | 0.285 | -0.005 |
| c51 | 1e7 | 0.262 | 0.307 | 0.045 |
| rainbow | 5e4 | 0.231 | 0.233 | 0.002 |
| rainbow | 3e6 | 0.319 | 0.443 | 0.124 |
| rainbow | 5e6 | 0.293 | 0.238 | -0.055 |
| rainbow | 8e6 | 0.310 | 0.370 | 0.060 |
| rainbow | 1e7 | 0.321 | 0.368 | 0.047 |
| vsarsa | 5e4 | 0.120 | 0.121 | 0.000 |
| vsarsa | 3e6 | 0.587 | 0.591 | 0.004 |
| vsarsa | 5e6 | 0.361 | 0.361 | -0.000 |
| vsarsa | 8e6 | 0.856 | 0.858 | 0.002 |
| vsarsa | 1e7 | 0.995 | 0.999 | 0.003 |
| vqn | 5e4 | 0.264 | 0.264 | 0.000 |
| vqn | 3e6 | 0.493 | 0.492 | -0.000 |
| vqn | 5e6 | 0.648 | 0.651 | 0.004 |
| vqn | 8e6 | 1.000 | 0.999 | -0.001 |
| vqn | 1e7 | 1.000 | 0.998 | -0.002 |
| ppo | 5e4 | 0.263 | 0.334 | 0.071 |
| ppo | 3e6 | 0.270 | 0.313 | 0.043 |
| ppo | 5e6 | 0.245 | 0.297 | 0.052 |
| ppo | 8e6 | 0.272 | 0.268 | -0.004 |
| ppo | 1e7 | 0.286 | 0.267 | -0.019 |

## E.2    Breakout

Table 6: Table of $\mathcal{R}$ values for Breakout.

| Algorithm | Frames | Original | Intervened | Normalized |
|---|---|---|---|---|
| a2c | 5e4 | 0.474 | 0.472 | -0.002 |
| a2c | 3e6 | 0.762 | 0.637 | -0.125 |
| a2c | 5e6 | 0.746 | 0.627 | -0.119 |
| a2c | 8e6 | 0.764 | 0.654 | -0.110 |
| a2c | 1e7 | 0.733 | 0.666 | -0.066 |
| dqn | 5e4 | 1.000 | 0.999 | -0.001 |
| dqn | 3e6 | 0.691 | 0.593 | -0.098 |
| dqn | 5e6 | 0.588 | 0.611 | 0.023 |
| dqn | 8e6 | 0.646 | 0.612 | -0.034 |
| dqn | 1e7 | 0.626 | 0.617 | -0.009 |
| ddqn | 5e4 | 0.995 | 0.999 | 0.004 |
| ddqn | 3e6 | 0.666 | 0.607 | -0.059 |
| ddqn | 5e6 | 0.594 | 0.591 | -0.003 |
| ddqn | 8e6 | 0.586 | 0.617 | 0.031 |
| ddqn | 1e7 | 0.570 | 0.615 | 0.045 |
| c51 | 5e4 | 0.991 | 0.999 | 0.008 |
| c51 | 3e6 | 0.549 | 0.576 | 0.027 |
| c51 | 5e6 | 0.525 | 0.589 | 0.064 |
| c51 | 8e6 | 0.530 | 0.558 | 0.027 |
| c51 | 1e7 | 0.549 | 0.563 | 0.015 |
| rainbow | 5e4 | 0.467 | 0.470 | 0.003 |
| rainbow | 3e6 | 0.490 | 0.524 | 0.035 |
| rainbow | 5e6 | 0.516 | 0.494 | -0.022 |
| rainbow | 8e6 | 0.546 | 0.490 | -0.056 |
| rainbow | 1e7 | 0.512 | 0.495 | -0.017 |
| vsarsa | 5e4 | 0.429 | 0.429 | 0.000 |
| vsarsa | 3e6 | 1.000 | 0.999 | -0.001 |
| vsarsa | 5e6 | 1.000 | 0.999 | -0.001 |
| vsarsa | 8e6 | 1.000 | 0.999 | -0.001 |
| vsarsa | 1e7 | 1.000 | 0.999 | -0.001 |
| vqn | 5e4 | 0.540 | 0.537 | -0.003 |
| vqn | 3e6 | 0.729 | 0.840 | 0.111 |
| vqn | 5e6 | 0.717 | 0.793 | 0.076 |
| vqn | 8e6 | 0.623 | 0.806 | 0.184 |
| vqn | 1e7 | 0.633 | 0.680 | 0.047 |
| ppo | 5e4 | 0.480 | 0.507 | 0.027 |
| ppo | 3e6 | 0.511 | 0.510 | -0.001 |
| ppo | 5e6 | 0.529 | 0.513 | -0.016 |
| ppo | 8e6 | 0.485 | 0.508 | 0.023 |
| ppo | 1e7 | 0.497 | 0.509 | 0.012 |

### E.3 Space Invaders

Table 7: Table of $\mathcal{R}$ values for Space Invaders.

| Algorithm | Frames | Original | Intervened | Normalized |
|-----------|--------|----------|------------|------------|
| a2c | 5e4 | 0.350 | 0.348 | -0.002 |
| a2c | 3e6 | 0.451 | 0.420 | -0.030 |
| a2c | 5e6 | 0.477 | 0.412 | -0.065 |
| a2c | 8e6 | 0.438 | 0.389 | -0.050 |
| a2c | 1e7 | 0.435 | 0.379 | -0.056 |
| dqn | 5e4 | 1.000 | 0.999 | -0.001 |
| dqn | 3e6 | 0.466 | 0.328 | -0.138 |
| dqn | 5e6 | 0.426 | 0.408 | -0.018 |
| dqn | 8e6 | 0.427 | 0.450 | 0.022 |
| dqn | 1e7 | 0.465 | 0.358 | -0.107 |
| ddqn | 5e4 | 1.000 | 0.999 | -0.001 |
| ddqn | 3e6 | 0.510 | 0.504 | -0.006 |
| ddqn | 5e6 | 0.492 | 0.547 | 0.056 |
| ddqn | 8e6 | 0.505 | 0.437 | -0.067 |
| ddqn | 1e7 | 0.486 | 0.423 | -0.064 |
| c51 | 5e4 | 1.000 | 0.999 | -0.001 |
| c51 | 3e6 | 0.418 | 0.335 | -0.083 |
| c51 | 5e6 | 0.438 | 0.382 | -0.056 |
| c51 | 8e6 | 0.509 | 0.439 | -0.070 |
| c51 | 1e7 | 0.512 | 0.468 | -0.043 |
| rainbow | 5e4 | 0.340 | 0.350 | 0.010 |
| rainbow | 3e6 | 0.594 | 0.671 | 0.077 |
| rainbow | 5e6 | 0.534 | 0.676 | 0.142 |
| rainbow | 8e6 | 0.543 | 0.641 | 0.098 |
| rainbow | 1e7 | 0.511 | 0.636 | 0.125 |
| vsarsa | 5e4 | 0.264 | 0.264 | 0.000 |
| vsarsa | 3e6 | 0.852 | 0.858 | 0.006 |
| vsarsa | 5e6 | 0.594 | 0.590 | -0.004 |
| vsarsa | 8e6 | 0.600 | 0.706 | 0.106 |
| vsarsa | 1e7 | 0.709 | 0.775 | 0.067 |
| vqn | 5e4 | 0.394 | 0.444 | 0.049 |
| vqn | 3e6 | 0.733 | 0.722 | -0.010 |
| vqn | 5e6 | 0.898 | 0.988 | 0.090 |
| vqn | 8e6 | 0.665 | 0.802 | 0.138 |
| vqn | 1e7 | 0.811 | 0.937 | 0.127 |
| ppo | 5e4 | 0.355 | 0.354 | -0.001 |
| ppo | 3e6 | 0.380 | 0.363 | -0.017 |
| ppo | 5e6 | 0.374 | 0.382 | 0.008 |
| ppo | 8e6 | 0.421 | 0.345 | -0.076 |
| ppo | 1e7 | 0.422 | 0.324 | -0.098 |

