# OpenReview forum: "Measuring Offline Robustness in Reinforcement Learning"
_TMLR — Rejected by TMLR_

### Review · Reviewer_KRkm · 2022-11-01

**Summary Of Contributions:**

The authors approach RL from the perspective of explainable AI, and want to devise some measures of reliability and reproducibility of an RL algorithm.

The robustness of an RL pipeline is defined as how much change occurs to the learned agent of the RL pipeline as its inputs are changed. For the course of the paper, the change in inputs is fairly simple: RL tends to naturally include random sampling or an RNG in some way, and the authors use different RNG seeds to generate different RL agents.

This leaves measuring how robust the agent is. The paper takes the position that maximizing reward is not a sufficient measure of RL agents - it is possible for there to be many good agents with very different behavior, and in a real-world deployment setting, users will care more that the actions of the agent are stable, rather than the end reward. So, the authors propose measure robustness as the divergence between action distributions of different agents $A_k$, from different runs of the same RL algorithm, given the same state $s$. Using this measure, they compare the robustness of different RL agents at different steps of model training. They find that in general, agents early in training tend to be robust (similar actions with poor performance), and later in training agents would sometimes become less robust depending on algorithm, with Rainbow keeping the most robustness while still having high performance.

**Audience:**

Yes

**Claims And Evidence:**

Yes

**Requested Changes:**

Some discussion of ways to decide the tradeoff between performance and robustness would be interesting, I did not see any discussion or suggested guidance on what to do there.

Different variations / perturbations to the RL pipeline instead of just random seed.

**Strengths And Weaknesses:**

The paper is generally well written, although I did find it confusing that the states $s$ used to estimate robustness were described as interventions. By my understanding, the states $s$ are primarily treated as an offline dataset of states that represent the parts of the state space that we want to judge our models against. Once queried, the actions from the agents are never used to rollout new episodes in the environment - so it doesn't really feel like an intervention. It feels more like an offline measure.

At a higher level, although the definition of $\mathcal{R}$ seems reasonable, I do have objections to the choice of randomness used to generate different agents. I believe the authors only use random seeds between RL algorithms. In a true practical setting, it's possible for someone to fix the seed of the RL training before running it if they want to get reproducible behavior. In fact, this is exactly what I assume the authors did when generating experiments for this paper!

Although random seed is a potential source of randomness, it seems like the paper would be significantly more useful if it compared robustness across a different variation for each algorithm (such as differing hyperparameters, or incrementally including extra episodes in training or not). My concern is that the robustness against just random seed would not be too useful. In my experience real world RL problems tend to be concerned with robustness of an agent as it receives additional training data, rather than robustness across different random seeds.

---

> ### Author Response · Authors · 2022-12-02
> **Comments discussing offline data, random seeds, sources of variability, and performance v. robustness**
>
> “Intervention” vs. “offline dataset of states” — You suggest that what are termed “interventions” in the submitted draft would better be termed “an offline dataset of states that represent parts of the state space that we want to judge our models against.” We agree. Yours is a more generalized characterization of our goals in devising a measure of IR. We have updated the text of the paper and the discussion to correspond to this notion.
>
> Fixing the random seed — You note that “it's possible for someone to fix the seed of the RL training before running it if they want to get reproducible behavior.” That is correct, but may not work well in many practical settings. For example, in many deployed applications, practitioners would want to periodically deploy models retrained on the newest data acquired. Our results show that the retrained agent will likely take different actions from the original agent even if the new data is drawn from exactly the same distribution as the original data, due to the variability that comes from other incidental features of training (which are controlled by the random seed). IR is the measure of the variability that comes from incidental features exclusively (e.g., the particular data sample), not from intentional changes such as different hyperparameters or different training algorithms.
>
> Additional sources of variability — You note that “the paper would be significantly more useful if it compared robustness across a different variation for each algorithm (such as differing hyperparameters, or incrementally including extra episodes in training or not).” These other sources of variability would certainly be interesting to study, and the robustness measure that we propose could be used to characterize the variability in actions due to these sources. In any revised version, we will include a discussion of this (and other) uses of the measure. The aim of the existing experiments is to demonstrate how ubiquitous such variability is in RL even when the differences among the pipeline instances would not be expected to produce differences in policies. We believe that this makes the strongest and simplest case that a measure of robustness is useful and important for studying RL algorithms.
>
> Discussion of performance/robustness tradeoff — You request “[s]ome discussion of ways to decide the tradeoff between performance and robustness.” You also note that “The paper takes the position that maximizing reward is not a sufficient measure of RL agents…and in a real-world deployment setting, users will care more that the actions of the agent are stable, rather than the end reward.” We did not intend to imply that robustness is more important than the final return, nor that there is a necessary tradeoff between robustness and performance. However, this issue is clearly central and important to discuss. We have added such a discussion and will continue to emphasize these points in any revised version.

---

> > ### Comment · Reviewer_KRkm · 2022-12-09
> > **Reply**
> >
> > The author's reply to the random seed comment:
> >
> > "For example, in many deployed applications, practitioners would want to periodically deploy models retrained on the newest data acquired. Our results show that the retrained agent will likely take different actions from the original agent even if the new data is drawn from exactly the same distribution as the original data, due to the variability that comes from other incidental features of training (which are controlled by the random seed)."
> >
> > Yes, this is something I noted in the next section, when I said,
> >
> > "the paper would be significantly more useful if it compared robustness across different variations (such as [...] incrementally including extra episodes in training or not)"
> >
> > The authors experiments are not based on this practical setting proposed as an example, they are based on a static algorithm, static hyperparams, a static dataset, and a static random seed. You can create different learning behavior by perturbing any one of those choices - but the fact this is possible does not change the reality that the paper focuses on perturbing the entire algorithm or the random seed and not the other two factors. The fact we can construct scenarios where random seed is fixed and RL performance changes does not mean the paper demonstrates IR will apply well when that is done.
> >
> > If the aim of the existing experiments is to demonstrate "how ubiquitous such variability is in RL even when the differences among the pipeline instances would not be expected to produce differences in policies.", what is the distinction between this work and papers like Reinforcement Learning that Matters which also demonstrate differences based on random seed or code implementation chosen? It seems like the goal of the experiments should be demonstrating the usefulness of IR in making actionable decisions, rather than showing the IR score is greater than 0.

---

### Review · Reviewer_dDWT · 2022-11-07

**Summary Of Contributions:**

This paper defines _interventional robustness_ as a measure on the consistency of action selection across multiple trained agents under the same training pipeline. The motivation for this measure is as a signal for agent robustness to perturbations and/or interventions in the training process.
The authors validate their proposed measure by evaluating a number of different RL algorithms when training a few Atari games, and evaluating their consistency when performing "interventions" on these games.

**Audience:**

Yes

**Claims And Evidence:**

No

**Requested Changes:**

I like the general idea of the paper, but I think it needs to be made more convincing. I am not convinced looking at differences at the action-level is all that meaningful. Although the authors attempt to argue for this with Atari games, these are agents that are not meant to be deployed and it is not clear that any conclusions will necessarily hold for other agents or environments.

I would suggest the authors evaluate on a more varied set of environments, and ideally some that are closer to real-world problems (the [Balloon Learning Environment](https://github.com/google/balloon-learning-environment) could be a possibility).

More importantly, I strongly believe a notion of IR needs to be higher-level than individual action differences, as these can be quite fickle and fail to capture meaningful behavioural differences.

## Minor comments/suggestions
1. In algorithm 1, in the innermost for loop I think it should be $\mathbf{a_k}$ instead of $A_k$.
1. In Algorithm 3, why do you need agents $G_1,\ldots,G_n$ as inputs? It seems they are not used.
1. In Figure 4 it seems the performance column is showing the performance with the original game (e.g. without intervention). It would be more useful to show how performance is affected by interventions.

**Strengths And Weaknesses:**

# Strengths
I think the idea has merit, and indeed robustness and consistency are two desirable properties for any agent which will be deployed. I think the idea of interventions is a good one and such a concept seems to be useful for measuring general robustness and consistency.
I also like the visualizations in Figures 3 and 4, and feel they are effective at conveying the different IR measures for the different algorithms.
The paper was also well written and easy to follow.

# Weaknesses
## Action differences
My main complaint about this paper is that I feel it fails to deliver convincing arguments for the nice high-level motivation laid out. While behavioural consistency is certainly something we should ask of our RL agents, I don't think this consistency is reasonable at the primitive action level. The authors hint at this a little towards the end of page 2 ("It is, of course, possible that the policies ... could be non-robust at the level of individual actions, but still robust in terms of higher-level strategies.") but argue against considering higher-level policies, saying "it seems implausible since the high-level strategies could easily be subtly different". I disagree with this, as I think high-level policies are actually what matter. Some of the examples used throughout the paper (stopping at a stop sign) are actually using high-level actions (in a realistic scenario it wouldn't consist of a single "stop" action, but a series of complex control sequences).

## Evaluation
Although I appreciate the evaluation on a standard benchmark (ALE), I don't feel it is sufficient to be convincing about the merits and usefulness of the proposed metric.

### Atari
Atari agents were never meant to be "deployed" or "interpretable", so it is not clear what the takeaways are from the experiments. Does the fact that Rainbow seems to have higher IR than DQN mean this will always be the case, especially in more real-world domains? It is not clear.

### Interventions
The interventions are quite artificial and limited to pixel alterations. It is unclear whether these interventions can tell us much about general-form interventions. I would suggest the authors consider Atari modes (see [Jesse Farebrother, Marlos C. Machado, Michael Bowling. Generalization and Regularization in DQN](https://arxiv.org/abs/1810.00123).

More generally, it would be better to evaluate other, non pixel-based, forms of intervention. Perhaps some MuJoCo/Brax tasks? This also raises the question of how one would measure IR in continuous action spaces.

### Stochasticity
Although not specified, I believe the authors are running the Atari games with deterministic transitions. Stochastic transitions can be added by using sticky actions (see [Revisiting the arcade learning environment: evaluation protocols and open problems for general agents](https://www.jair.org/index.php/jair/article/view/11182/26388)), and I feel would be a better way of evaluating the type of agent robustness the authors are interested in.

In section 3.3 the authors state they are "sampling a set of states from a trajectory produced by one additional agent." Wouldn't it make more sense to sample from a few trajectories? Otherwise, if the environment is stochastic, this could lead to biased samples. In the current evaluations, this point would be moot, which is why I suggest evaluating with stochastic environments.

### Conclusions
The authors conclude with: "we demonstrate that agent performance is not strongly predictive of IR." This is only for the specific form of IR introduced in the paper and for the few games run on a few agents. I am not sure this conclusion holds in general.

## Missing literature
There are some papers that I feel should be discussed, as they are quite related:
1. The notion of individual actions moving around is related to policy churn: [Tom Schaul, André Barreto, John Quan, Georg Ostrovski. The Phenomenon of Policy Churn. NeurIPS 2022](https://arxiv.org/abs/2206.00730)
1. One aspect that is not at all discussed is the notion that some actions might be _equivalent_, and so switching from one to the other doesn't matter at all (e.g. they are effectively identical). The current notion of IR would not capture this. I suggest the authors look at things like state-action metrics (e.g. [Taylor, Precup, Panangaden. Bounding Performance Loss in Approximate MDP Homomorphisms. NeurIPS 2008](https://proceedings.neurips.cc/paper/2008/hash/6602294be910b1e3c4571bd98c4d5484-Abstract.html) and related works).

---

> ### Author Response · Authors · 2022-12-02
> **Comments discussing actions v. behaviors, the use of Atari, pixel-level interventions, stochasticity, related literature, and performance plots**
>
> Robustness of actions vs. behaviors — You note that “[w]hile behavioural consistency is certainly something we should ask of our RL agents, I don't think this consistency is reasonable at the primitive action level.” This raises at least two issues. First, as you imply, robustness (or consistency) can be measured across a range of scales, from primitive actions to high-level behaviors. Research in RL has long considered how to aggregate actions into higher-level behaviors (often called “macro-actions” or “options”). The measure we define (IR) can be applied to any discrete set of actions, regardless of whether they are the primitive actions or some higher-level option. We chose to demonstrate the application of IR using primitive actions, but the measure is in no way limited to that level. That said, there is no current consensus within RL about how to reliably discover higher-level actions, so it would be challenging to demonstrate interventional robustness using such higher-level actions. We have modified the initial description of IR to emphasize that it is defined for any set of actions, whether they be primitive or higher-level, and describe how it might be used given a particular set of higher-level actions. Second, our aim with this paper was to define an intuitive measure of IR, demonstrate its application, and evaluate whether it measures something independent of performance. You note that it may not be reasonable to expect robustness at the primitive action level, and we agree. It may or may not be reasonable to expect high IR for particular applications. Instead, our goal is whether our proposed measure is informative, as well as intuitive and relatively simple to apply. We will emphasize these goals more clearly in the revised draft.
>
> Use of Atari — You note that “I don't feel [ALE] is sufficient to be convincing about the merits and usefulness of the proposed metric… Atari agents were never meant to be ‘deployed’ or ‘interpretable’, so it is not clear what the takeaways are from the experiments.” Overall, the goal of the paper was to propose an intuitive measure of robustness and demonstrate its use. We chose the intervenable Atari environment because it is a widely used benchmark, and it was sufficient for the research questions we had. Specifically, we were interested in comparing action variability among RL agents with different training data (via different random seeds). Further, we were interested in seeing if performance was highly correlated with IR (if it was, why not just use performance in place of IR?). The Atari environment is complex enough to show that agents could have high variability and that performance and IR are not always correlated.
>
> Pixel-level interventions — You state that “The interventions are quite artificial and limited to pixel alterations.” First, we are not using pixel-level interventions. Rather, we are using an intervenable Atari environment, so the interventions affect high-level aspects of the state of the underlying simulator. For example, removing an enemy in Space Invaders actually removes the enemy from the game. We make this distinction clearer in the updated version of the paper.
>
> Stochasticity — You suggest using stochastic rather than deterministic transitions. We suspect that using an environment with stochastic actions would have relatively little effect on the high-level outcomes of our experiments, but this is an empirical question worth investigating.  Since ALE is a benchmark that does not include stochastic transitions, conforming to the benchmark, particularly in the original, non-intervened, in-distribution case, seems most appropriate. However, if investigating environments with stochastic transitions seems necessary, we would be willing to include further experiments in a later revised version of the paper.
>
> Writing and related literature — We will make the claims you mention clearer and/or modify them. For example, changing “agent performance is not strongly predictive of IR” to “agent performance is not necessarily strongly predictive of IR” is more accurate and in line with what we intended. As for the related literature, we have included a discussion of policy churn and will include a discussion of action equivalence. We also plan to discuss action equivalence in the future work as a potential improvement to IR.
>
> Performance plots — Since the focus of the paper is on the variability of actions and not performance, we reported the IR for the intervened environments and just the on-policy performance. (Getting the performance in the intervened environments would require the full trajectory, not just samples, so we could not get this trivially.) Exploring the variability of performance rather than actions is an interesting empirical question, though not the one we explore in this paper.

---

> > ### Comment · Reviewer_dDWT · 2022-12-12
> > **Reviewer response**
> >
> > Thank you for your responses. As you stated in your response, the main point of the paper "is whether our proposed measure is informative, as well as intuitive and relatively simple to apply". I agree it is simple to apply, somewhat agree with it being intuitive, but am not convinced of it being informative.
> >
> > To be convincing in this regard requires empirical arguments, which I feel is not the case yet (the reasons for this were stated in my original review). You agree that "that it may not be reasonable to expect robustness at the primitive action level", so if that is the only empirical evaluation conducted, how is a reader to determine whether the proposed measure is informative?
> >
> > Regarding stochasticity, many recent benchmarks use the ALE with stochastic actions, so it is relatively common to use. Further, your paper is not using benchmarks as a way to determine agent performance, but as a _test bed_ for determining whether your proposed measure is useful. So I think the focus should not be on whether you are using environments in a "standard" way or not, but rather, whether it is useful towards your purpose of being convincing. You are already modifying the benchmark by adding interventions, so I don't think adding stochasticity is unreasonable.
> >
> > But at a higher level, I think going beyond ALE is more important, as otherwise I don't see convincing enough empirical arguments for the usefuleness/validity of the proposed measure.

---

### Review · Reviewer_cE6r · 2022-11-18

**Summary Of Contributions:**

The authors introduce the novel concept of "interventional robustness", which they define to be the extent to which an agent's behaviour under environmental interventions varies with incidental properties of the training process such as parameter initializations and random number seed, which they refer to as a pipeline. Specifically, the authors modify the environment in some a priori reasonable ways (e.g. removing the cover in Space Invaders) and take the JS divergence between policies sampled from the same pipeline. Doing this on a per-state and per-intervention basis yields a 2D grid of IR values which, the authors claim, can be used for evaluating which algorithm/pipeline to choose for some application.

**Audience:**

Yes

**Claims And Evidence:**

No

**Requested Changes:**

As mentioned in the previous section, I don't believe the authors currently present any evidence for their claims. Concretely, I'd like to see IR used to select a pipeline, and show that's better than alternative metrics in terms of e.g CE consistency, user rated interpretability, etc.

I'd also like to see the process by which IR can be used to select amongst pipelines explicated, and the use of sample-based JS divergence either justified or changed.

**Strengths And Weaknesses:**

The presentation is quite strong, particularly the RL training pipelines in Figure 1. The idea of thinking holistically about all of the incidental variation involved in training an agent with RL is a good one, and I hope the authors continue pursuing this general research direction.

1) Construct Validity

That said, I have serious concerns about the construct validity of IR as defined here. They claim that "pipelines with high IR have a variety of useful properties. They behave similarly over time, fulfilling users’ expectations, and other software systems can rely on their outputs. Robust pipelines are good candidates for CE, and their previous explanations remain relevant over time.", but I don't see evidence for any of this. I have a vague sense about why we should care about the pipeline over specific instances, but even this could use a concrete experiment exhibiting where the latter could go wrong. But even if you provide this evidence, it still wouldn't be clear why the variation between seeds and post-intervention policies is what we should care about. Why shouldn't we just care about the extent to which performance degrades in the face of these interventions? Or the extent to which seeds are amenable to CE?

Indeed, my intuition is that a high IR might be actively harmful. For environments with multiple policies at a given level of performance, IR rewards pipelines that reliably latch on to one specific policy. A great deal of work in RL is actively involved in preventing this, as such a pipeline must be biased in manner orthogonal to performance, and e.g. an ensemble of agents from such a pipeline couldn't provide coverage of the solution space or meaningful uncertainty estimates. This intuition could, of course, be proven wrong given empirical evidence, but your current experiments don't weigh in on this core issue of construct validity.

Your experiments appear to be purely descriptive: you ran several pipelines across several environments and show the resulting IR grids over time. But this doesn't address whether or not IR is useful for selecting pipelines for CE, or any of the other use cases mentioned throughout the paper.

2) Using IR for pipeline selection

To even use IR for pipeline selection, it must be clear how two pipelines can be evaluated against each other. Currently each pipeline yields a checkpoint X state X invention multi-dimentional array of numbers. Should we be taking the average IR across all of these dimensions? The minimum? I imagine this might vary across different use cases, but showing at least a few concrete examples of pipeline selection seems necessary.

3) IR calculation

The choice of JS divergence on action samples seems problematic. Many of the policies examined here are deterministic, so the JS divergence is likely to be undefined. I imagine a sample-based approximation still yields finite results, but more care should be given to their interpretation. And why are action samples being used in the first place, when the explicit policy is known?

---

> ### Author Response · Authors · 2022-12-02
> **Comments discussing construct validity, high v. low IR, experiments, pipeline selection, and JS divergence**
>
> Evidence for construct validity — You call attention to our summary of the benefits of pipelines with high IR and note that “I don't see evidence for any of this.” We believed that these claims followed logically from the definition of interventional robustness, but it is clear that those connections (and potential caveats) deserve far more careful explanation. In a revised version, we will substantially refine and expand this section to make those logical connections much clearer.
>
> Potential harm of high IR — You note that “my intuition is that a high IR might be actively harmful,” and go on to note that “[f]or environments with multiple policies at a given level of performance, IR rewards pipelines that reliably latch on to one specific policy.” The question of whether high IR might be harmful in some contexts is a fascinating one, and we appreciate you raising this question. It reinforces a basic point of the paper: IR is useful to measure. Our primary point in this paper is not that IR is universally beneficial or harmful, but that it can and should be measured. Indeed, examining the question of when high IR is useful is nearly impossible to study empirically without a quantitative measure of IR. In a revised draft, we will note a much wider range of questions which can be addressed with a quantitative measure of IR.
>
> Experiments and Pipeline Selection — You note that the experiments in the paper are “purely descriptive” and that they don’t “address whether or not IR is useful for selecting pipelines for CE, or any of the other use cases mentioned throughout the paper.” Our goal in the paper was to define an intuitive measure of interventional robustness, demonstrate its application, and provide evidence that it does not necessarily increase monotonically with the amount of training and is not necessarily correlated with on-policy performance. Our goal in discussing CE and other potential applications was to demonstrate that robustness mattered, not that IR was a complete solution to these complex problems.
>
> In addition to clarifying our high-level motivation (see the “evidence for construct validity” section), we will include some example use cases for IR. For example, these could include cases where the average or minimum IR is desirable, as well as cases where globally low IR might be desirable (see the “potential harm of high IR” section).
>
> Use of IR for pipeline selection —  You point out that “[t]o even use IR for pipeline selection, it must be clear how two pipelines can be evaluated against each other.” As mentioned above, IR is not meant to be a simple measure that will completely determine the choice of pipeline. However, clearly more discussion is needed about how IR could be used as part of a more complex decision process. We will add such a discussion.
>
> Jensen-Shannon Divergence — You note that “The choice of JS divergence on action samples seems problematic. Many of the policies examined here are deterministic, so the JS divergence is likely to be undefined.” We chose bounded JS divergence in part because it can be modified to apply to both deterministic and stochastic policies, though we are open to other suggestions for similarity measures. The bound was derived from the reference “Divergence Measures Based on the Shannon Entropy” by Jianhua Lin. The exact calculation can be found in js_divergence.py in the attached code. For a deterministic policy, one action has a probability of 1.0, and the rest of the actions have a probability of 0.0. If every action from the group of agents is the same, then the bounded JSD value is 0. If they are all different, then the bounded JSD value is log2(10) for 10 agents.
>
> Samples of the stochastic policies were used so there would be a uniform interpretation of IR for both deterministic and stochastic policies and so that IR reflects both the similarity and the variability of action selection among agents. Reflecting the variability of action selection requires more than just comparing whether two agents have similar probabilities of action selection. For example, say we have two stochastic policies that select an action uniformly randomly: [0.2, 0.2, 0.2, 0.2, 0.2] and [0.2, 0.2, 0.2, 0.2, 0.2]. If we directly compared the distributions, then the JSD would be 0 since the distributions are the same. However, the actual actions that the two agents would take would be uniformly random (producing low IR by our interpretation), so we compare the similarity of sampled actions. We determined that drawing 30 samples would be accurate enough for our purposes. We are also open to suggestions for changes to the measure, as long as they retain the general properties of IR laid out in the paper.

---

> > ### Comment · Reviewer_cE6r · 2022-12-06
> > **JS Divergence between two identical distributions**
> >
> > Thank you for the detailed response.
> >
> > I believe your example comparing two identical uniform distributions illustrates my underlying point: if you want say that these two *identical* distributions yield high IR, then you can't also be trying to approximate the JS divergence, since all divergences would be minimized in this case. Rather, you seem to be using an artifact of your approximation method to measure something other than the JS divergence. I'm not sure what exactly you are trying to measure (mutual information?), but I'm confident that its not a divergence and should be fully explicated prior to publication.

---

> > ### Comment · Reviewer_cE6r · 2022-12-06
> > **IR is not self-evidently useful to measure**
> >
> > > You call attention to our summary of the benefits of pipelines with high IR and note that “I don't see evidence for any of this.” We believed that these claims followed logically from the definition of interventional robustness, but it is clear that those connections (and potential caveats) deserve far more careful explanation. In a revised version, we will substantially refine and expand this section to make those logical connections much clearer.
> >
> > You haven't shown that IR literally follows from the definition of interventional robustness (though a formal treatment showing this was somehow the case would be appreciated). And even if you did, then it would remain unclear why one should measure IR as opposed to interventional robustness directly. You appear to agree with me that you haven't shown any empirical evidence for IR's utility. I simply don't think better rhetoric is sufficient evidence since the utility of IR is the core contribution of this paper. I cannot support it's publication without at least some theoretical or empirical evidence that IR is worth measuring.
> >
> > > The question of whether high IR might be harmful in some contexts is a fascinating one, and we appreciate you raising this question. It reinforces a basic point of the paper: IR is useful to measure.
> >
> > This understands my point with raising this issue. If something might be beneficial, might be harmful, or might be neutral, then it raises the question as to how its measurement is intended to be used.

---

### Decision · Action_Editors · 2022-12-27

**Recommendation:** Reject

**Comment:**

The reviewers all found the idea of studying the incidental variations involved in training an RL agent to be interesting. However, they were also unanimous about the limitations of the precise construct proposed in the paper, namely interventional robustness (IR). Even after rebuttal and revision, it is not clear what IR is measuring, why, how it can be used and what would be the possible outcomes. They also raised concerns about the empirical evaluation (only varying seeds, evaluation not broad and thorough enough). The reviews and replies to the rebuttal provide more information.

**Audience:**

Yes

**Claims And Evidence:**

No, see below.